# Quality Assessment of Natural Juices and Consumer Preferences in the Range of Citrus Fruit Juices

Małgorzata Kowalska [1,*], Justyna Konopska [1], Melánia Feszterová [2,*], Anna Zbikowska [3] and Barbara Kowalska [3]

1. Department of Management and Product Quality, Faculty of Chemical Engineering and Commodity Science, Kazimierz Pulaski University of Technology and Humanities, 26-600 Radom, Poland
2. Department of Chemistry, Faculty of Natural Sciences and Informatics, Constantine the Philosopher University in Nitra, 949-01 Nitra, Slovakia
3. Department of Food Technology and Assessment, Institute of Food Sciences, Warsaw University of Life Sciences-SGGW (WULS-SGGW), 02-772 Warsaw, Poland
* Correspondence: mkowalska7@vp.pl (M.K.); mfeszterova@ukf.sk (M.F.); Tel.: +48-483617547 (M.K.); +421-903456414 (M.F.)

**Abstract:** The purpose of the study was to analyse and update consumers' changing preferences in the choice of citrus fruit juices and to evaluate the sensory and physicochemical characteristics of two kinds of juices: juice squeezed from raw fruit and a commercial juice indicated by respondents as best matching their preferences. The survey was conducted in the form of an online survey posted on app.ankieteo.pl. The survey was also sent via a link through social networks. A total of 862 people took part in the survey. Consumers are most likely to consume juices one to three times a week (28.3%). Orange juice was the most popular among respondents (52.4%). The main factors influencing decisions to purchase citrus fruit juices are the type of fruit from which the juice was made, the vitamin content and the product's price. In choosing juices, respondents were also guided by favourable health qualities and the presence of minerals. From the physicochemical determinations of orange juices obtained from a juicer and squeezer and commercial juice "O", it was found that the quality of commercial orange juice indicated by consumers in the survey is comparable to juices made with a squeezer or a juice.

**Keywords:** consumer preferences; citrus juices; vitamin C; glucose; fructose; quality

## 1. Introduction

The form, type and principles of nutrition significantly affect a person's well-being and health. Both citrus fruits and juices made from them have high nutritional value. They are mainly associated with their high vitamin C content. Still, they are also a source of many bioactive compounds such as carotenoids, flavonoids, and limonoids and contain vitamins such as vitamin B, thiamine, riboflavin, niacin, folic acid inositol, biotin, and choline [1].

The fruit industry is an important market segment. The early 1990s was a time when expensive imported juices were available on the market. However, like other food sectors, the juice sector has begun to grow through several technology modifications, increased assortment, and marketing efforts [2]. As a result, since 2010, the production of citrus fruit juices in Poland has remained above 100,000 tons per year, except in 2012, when it registered a lower production of 96,586 tons based on the IERiGŻ date 2011–2019 [3].

Juices can be produced at home and in small factories, but most are made on a large scale in industrial plants. Juice production is closely associated with the target product, i.e., whether the juice is clear, meth or puree [4].

The quality of juices is influenced, among other things by: the quality of the raw material used in juice production, the degree of maturity of the fruit, the method of harvesting, and postharvest operations [5]. Fruit maturity is determined by the percentage

of juice in the fruit, total soluble solids, and the colour of the fruit. Size, shape, absence of blemishes, deformations, and other damage, and absence of pathogens and pests are also important parameters in determining fruit quality [6]. An important factor is the growing conditions of the raw material from which the juice is produced. Growing conditions affect the vitamin C content the fruit. Temperature and the amount of light provided during vegetation affect its content; the less light during growth, the less vitamin C in the plant [7]. This vitamin is the component of citrus juices most recognised and associated with.

Citrus juices, like the fruits, are rich in many nutrients, including vitamin C, and also contain various bioactive components such as (poly)phenols [8]. According to Berk [9]. (2016), citrus juices are a fairly good source of minerals, particularly potassium. Typical values of potassium content per 100 g are 150 mg in grapefruit juice and 200 mg in orange juice. In the other hand, citrus juices are very low in calcium content. Calcium content and vitamin D are added to some fortified citrus juices to make them more nutritionally attractive as a breakfast drink for young children. Citrus fruits and their juices are among the recommended folic acid sources in most dietetic guides. Vitamin C is important, mainly as a cofactor of several enzymes responsible for collagen biosynthesis. Citrus juice seems to be an ideal medium for iron-supplementing food for small children [10]. Marcus [11] states that citrus fruit and citrus juice impart acid to formulations and recipes that help to add brightness and heighten the perception of other basic tastes. Citrus juice helps to balance flavour, highlight natural flavours, preserve colour and prevent oxidation. Among other things, citrus and tropical juices are increasingly used as ingredients in soft drinks, flavoured waters and smoothies.

According to Farris [12], vitamin C cannot be synthesised in humans because we lost the ability to produce L-gulono-lactone oxidase, the ingredient (enzyme) responsible for its production. Thus, this vitamin must be obtained from dietary sources: citrus fruits and leafy green vegetables. In addition, vitamin C increases the body's immunity by inhibiting the proliferation of viruses [13], which is very important in these times of pandemics. It has also been proven that the supply of vitamin C in higher doses increases the body's immunity, more specifically, the number of leukocytes [14].

This study aimed to investigate consumer preferences in choosing citrus fruit juices. In the survey, respondents were asked, among other questions, about the frequency of juice consumption, the most preferred flavour, the type of juice, and what influenced the decision to purchase these products. Following a questionnaire survey that yielded information on the most popular commercial orange juice, a comparative evaluation of the juice's physicochemical and sensory characteristics was made against fresh juice squeezed from raw fruit using a juicer or squeezer prepared in a sensory laboratory.

## 2. Materials and Methods

### 2.1. Questionnaire Design

A survey on consumer preferences for juices from citrus fruits was conducted in late April and early May 2022. The survey was posted on app.ankieteo.pl and was also distributed via a link through social networks. 862 people took part in the survey. The survey form consisted of 16 questions. The first question asked about the frequency of consumption of citrus fruit juices. If a respondent declared that he or she consumed juices infrequently, occasionally, or not at all, he or she was asked in the second question to give a reason that influenced this. Further questions addressed issues such as: "What are the specific reasons for consuming citrus fruit juices?", "What flavour is most often chosen?", and "What type of juices are most often selected—due to the form of production, appearance and texture, and condition of preservation?". In addition, respondents were asked to indicate to what extent the given factors are essential when buying citrus fruit juices, what brand of juice they often buy and why, and whether they happened to choose something other than juice, nectar, or beverage. The last four questions focused on age, education, gender, and the size of the locality of residence.

### 2.2. Statistical Analysis

The basic test that was used in the statistical analyses was the chi-square test for the variables' independence. It was mainly used for questions built on nominal scales. Coefficients based on Kramer's V test were used to determine the strength of the relationship. Analysis by the chi-square test is accurate when none of the theoretical abundances are less than unity and when no more than 20% of the theoretical abundances are less than 5. Therefore, for each analysis by the chi-square test, additional tests were performed, which were conducted with especially small samples. These were tests performed with the following methods: exact or Monte Carlo. The estimated test probability "$p$" indicates whether the analysed relationship is statistically significant. The significance of "$p$" and Kramer's V is determined by the result of the chi-square test.

When the variables were ordinal, the coefficients used were Kendall's Tau-b for tables with the same number of columns and rows and Kendall's Tau-c for tables with different numbers of columns and rows. The values of the coefficients can take, as in the case of the Phi measure, negative results. Thus, they allow us to determine not only the strength but also the direction of the correlation. Under each cross table, among other things, you will find the value of the coefficient and statistical significance "$p$". In addition, the "$p$" value is calculated using the Monte Carlo method, which is also marked with a (c). If the analyses showed that there were no statistically significant correlations using Kendall's Tau-b and Tau-c coefficients, then the correlation was checked based on the chi-square test and the corresponding symmetric measure informing the strength of the relationship. This made it possible to detect a possible relationship that is not monotonic in nature.

Measures of relationship strength for the aforementioned coefficients range from 0 to 1, with a higher coefficient value indicating stronger dependence/correlation. As mentioned above, the results obtained can also take negative values (except for Kramer's V measure), which indicates the direction of dependence, but the interpretation of the strength of the relationship is similar.

The Monte Carlo method is mostly based on a sample of 10,000 tables with a starting number of a random number generator of 2,000,000.

The analysis was performed using the IBM SPSS 26.0 package with the Exact Tests module. All relationships/correlations/differences are statistically significant when $p \leq 0.05$.

In the second part of the work, after analysing the questionnaire results, commercial, orange-flavoured juice and juices made independently from oranges from two different countries of origin from, Spain and Egypt, (that were available on the market at that time in the country) were subjected to sensory and physicochemical analysis.

The orange juices were squeezed using a Kenwood citrus squeezer and a Philips juicer. Both squeezed juices were cloudy and contained pulp particles.

### 2.3. Sensory Evaluation

The juices were subjected for colour, aroma, taste, and appearance. They were evaluated immediately after squeezing, and juice "O" immediately after opening, at room temperature. Evaluators were given described samples of each juice and an evaluation sheet, which included organoleptic requirements for orange juice. Evaluators were asked to rate each characteristic on a scale from 1 to 5 where: 1—bad quality, 2—insufficient quality, 3—sufficient quality, 4—good quality, 5—very good quality.

General recruitment requirements given in standards [15–18] were used during the training of the test subjects. During the recruitment of the panel, attention was paid to the availability and willingness of the candidates to perform various tests over an extended period of time, as well as their motivations.

Preparation of the panel for sensory determinations and all qualification tests were carried out at the sensory analysis laboratory of Kazimierz Pulaski University of Technology and Humanities in Radom. The training lasted 20 h. During the training sessions, future assessors were introduced to various methods ranging from simple assignment methods, recognition methods, and difference detection methods. At first, single taste and smell

substances were used. Later, mixed samples, consisting of two or more ingredients in different proportions, were presented to the assessors for testing.

Checking the candidates' ability to correctly perceive colours was carried out using pseudo-chromatic arrays developed by Ishihara.

In the evaluation of the sense of taste, recruited subjects were asked to arrange and rank 7 taste samples differing in intensity. The test was conducted for sweet, sour, and bitter flavours. The concentrations of the primary flavours were developed based on [16]. The critical value of Spearman's rank correlation coefficient (translating into the ability to correctly rank the samples) was set at 0.7.

During the determinations, evaluators learned the citrus fragrances. In each session, 10 different substances were provided for testing. A total of two sessions were held. The percentage of correct responses was calculated based on the rating scale given in the standard [19], where individual responses were assigned the following number of points; 3 points—for correct identification or description of the most common associations, 2 points—for describing the sample with general terms, 1 point—for identification. It was assumed that those who did not score 65% in such tests did not qualify as assessors.

In our study, 8 people were qualified for the team, and after general training and qualification they were subjected to directional training (6 training sessions of 2 h each).

### 2.4. Physicochemical Evaluation

The following parameters were determined:

Density—using a kinematic viscosity meter from Anton Paar.

Sugar content—using a refractometer (company: ATAGO, Tokyo, Japan).

Dry matter content—using a moisture analyser manufactured by Radwag Wagi Elektroniczne Poland.

Vitamin C content. The test was conducted after direct squeezing of the juice and after pasteurisation of the juice. Vitamin C content was measured by the titration method [19].

Colour measurement—using a Konica Minolta Portable chroma meter-CR-410 colorimeter.

Efficiency, which was obtained by calculating what percentage of the weight of the orange was the weight of the squeezed juice.

### 2.5. Cost Analysis

The study analysed the cost of producing juices from natural fruits. The purpose of this analysis was to check if the cost of obtaining juices directly from natural fruits is comparable to the juice produced by the juice producer indicated in the survey. The analysis omitted the cost of energy consumption required to run the juicing machine. The analysis was conducted for 1 L of this product.

## 3. Results and Discussion

### 3.1. Analysis of Respondents' Answers

According to the recommendations of the Food and Nutrition Institute in Warsaw, Poland and the Pyramids of Healthy Eating and Physical Activity, vegetables and fruits should form the basis of the daily diet. People should consume at least five servings of vegetables and fruits per day as often as possible. Fruits play a significant role in a healthy balanced diet for a human and are so considered a vital food commodity around the world. Citrus is one of the most important fruit crops worldwide, with more than 130 million tons produced in 2017 [20]. Citrus fruit is one of the most essential commercial fruits in the world [21], it is the most economically relevant and extensively grown fruit tree crop in the world, and its fruits are an essential source of secondary metabolites for nutrition, health, and industrial applications [22]. Some authors recommend replacing fruit with one glass (200 $cm^3$) of juice, such as orange juice [23]. In the first question, respondents indicated how often they consume citrus fruit juices. The responses received from respondents indicated that they were most likely to consume juices one to three times a week (see Figure 1).

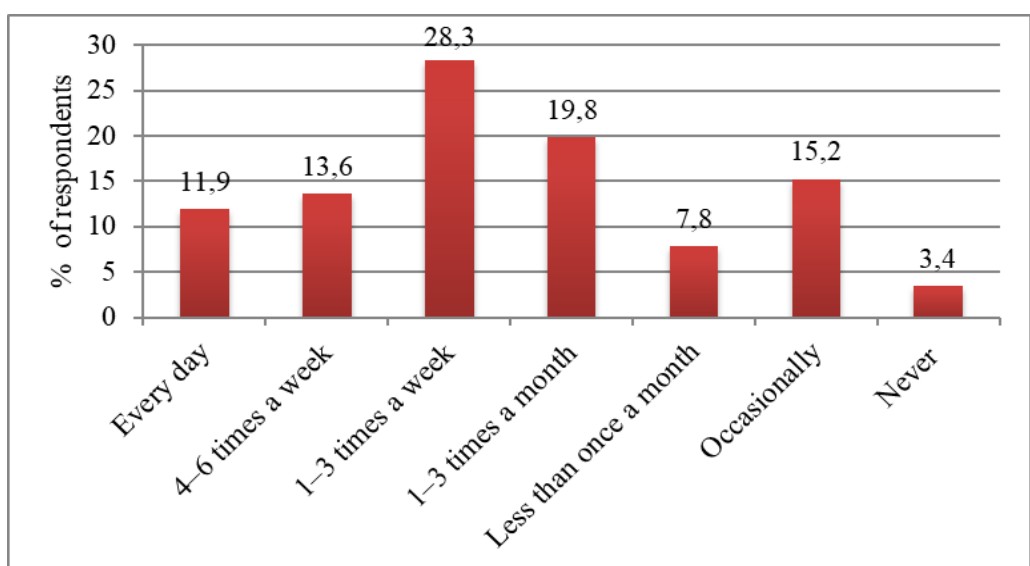

**Figure 1.** Frequency of consumption of citrus fruit juices by respondents.

Women's and men's preferences in this regard were at a similar level. Both women (28.4%) and men (28.1%) were most likely to consume juices one to three times a week. Men consumed juices with greater frequency, daily or almost daily. All age groups indicated in the survey (under 18, 18–24, 25–34, 35–44, 45–54, 55–64) were also most likely to consume juice one to three times a week, excluding the group of people 65 and older, who consumed them less frequently—the most indications in this group were recorded for frequency of consumption as one to three times a month (32.4%).

The statistical analysis carried out showed that gender, age, and size of the locality of residence had no statistically significant relationship with the frequency of consumption of citrus fruits juice. The following results were obtained sequentially for: gender of respondents for the analysed question (V Kramer = 0.097, $chi^2$ = 8.084, df = 6, $p$ = 0.232, p Monte Carlo = 0.229), age (V Kramer = 0.097, $chi^2$ = 48.447, df = 36, $p$ = 0.080, p Monte Carlo = 0.080), and size of locality of residence (V Kramer = 0.087, $chi^2$ = 32.810, df = 30, $p$ = 0.331, p Monte Carlo = 0.325). On the other hand, between the level of education and the frequency of consumption of citrus juices, a statistically significant relationship with insignificant strength of association was shown, but it was not monotonic (V Kramer = 0.113, $chi^2$ = 32.872, df = 18, $p$ = 0.017, p Monte Carlo = 0.017).

More than 25% of those surveyed indicated that they consumed juices less than once per month, occasionally, or not at all. In this case, respondents were asked to indicate the reason why they rarely consumed this type of juice.

The most common indication of respondents was: "I like juices, but I prefer to drink water" (45.1%) and "I prefer to eat fresh fruit" (38.8%) (see Figure 2). In all age groups, excluding the 45–54 age group and those aged 65 and over, respondents declared that they would rather choose water than juice. In contrast, the aforementioned two age groups were more interested in fresh fruit than in a product made from it. Respondents also indicated other reasons for not consuming citrus fruit juices. One of them that limited their choice in this direction was the high price of these products. The indication of price as one of the important determinants in the choice of juices was previously pointed out by Steenhuis et al. [24], who informed that price is an important factor in food purchases. In the work presented here, as expected, such an indication was characteristic of those with lower incomes. Another factor that limits the choice of citrus juices is an allergy to this type of fruit. The allergens in citrus fruits are proteins and glycoproteins, as well as fragrances, e.g., citral, citronellol, and limonene, which can cause contact dermatitis, gastrointestinal symptoms (including diarrhoea, vomiting, or constipation), as well as rhinitis and even asthma [25,26]. Nevertheless, it should be pointed out that for the entire surveyed group of

respondents, the important elements or reasons for which they made their choice when it came to citrus juices were health reasons and sensory appeal. An important piece of information is that some people (3.1%) were allergic to citrus.

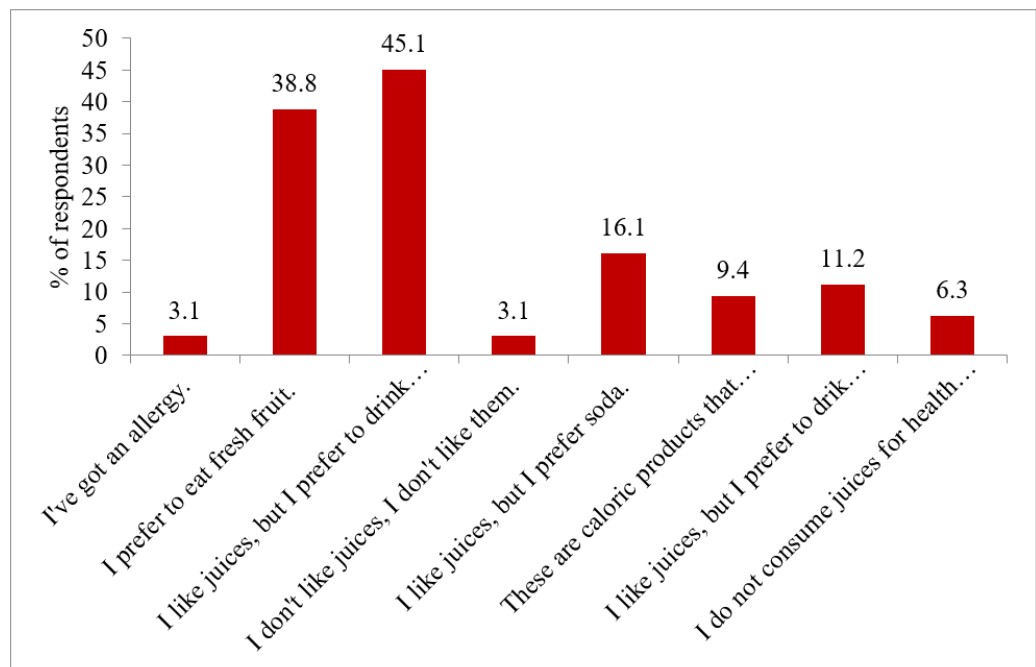

**Figure 2.** Reasons for infrequent consumption of citrus fruit juices by respondents.

Gender does not statistically significantly differentiate the reasons for not consuming citrus fruit juices. No statistically significant relationships were found between age and the listed reasons why respondents do not consume citrus fruit juices. Respondents from more urban areas were more likely to give other reasons for rarely consuming citrus fruit juices. Only the share of this one answer depended statistically significantly on place of residence. No statistically significant relationships were found between education level and the indication of reasons for not consuming citrus fruit juices.

The main motivation for juice consumption among respondents was taste sensation (62.1%). In second and third place, the most indications were related to health aspects associated with the consumption of this type of juice. The following answer: "they have beneficial effects on health" was indicated by 43.2% of respondents, while the answer: "they are a good source of vitamins and nutrients"—42.2% (see Figure 3).

Women were more likely than men to drink citrus juices because they like them, and according to them, juices are a good source of vitamins and nutrients. Men, on the other hand, were more likely to choose citrus juices because, according to them, juices quench thirst very well. The relationships described above are statistically significant and characterised by insignificant strengths of association. Younger respondents were more likely than older ones to drink citrus fruit juices because they like them. Only the proportion of this one response depends statistically significantly on age. The strength of the association is Kramer V = 0.238. The reasons why respondents do not drink citrus juices were not statistically significantly dependent on place of residence. Respondents with basic—vocational—education were less likely to indicate their choice of juice as an alternative to fresh fruit. Only the share of this one response is statistically significantly dependent on education level. The strength of the relationship is insignificant.

The largest share of the citrus juice trade is orange juice, followed by grapefruit, lemon, and lime. Other citrus fruit juices such as mandarin, tangerine, pomelo, and yuzu are traded in relatively small quantities. The most frequently indicated by respondents among citrus juices was orange juice (52.4%). Orange juice is the most popular juice in the world because it has aroma, attractive flavour and colour, and health benefits [27]. The orange

flavour is perhaps widely recognised and accepted in the food and beverage industry worldwide [28]. The flavour of fresh orange juice is due to the complex combination of a large number of volatile compounds, including esters, aldehydes, alcohols, ketones, and terpenes [29]. Another juice that respondents were also interested in and liked was lemon juice. Respondents were less interested in grapefruit juice and lime juice. The least attractive to respondents was pomelo-flavoured juice (see Figure 4).

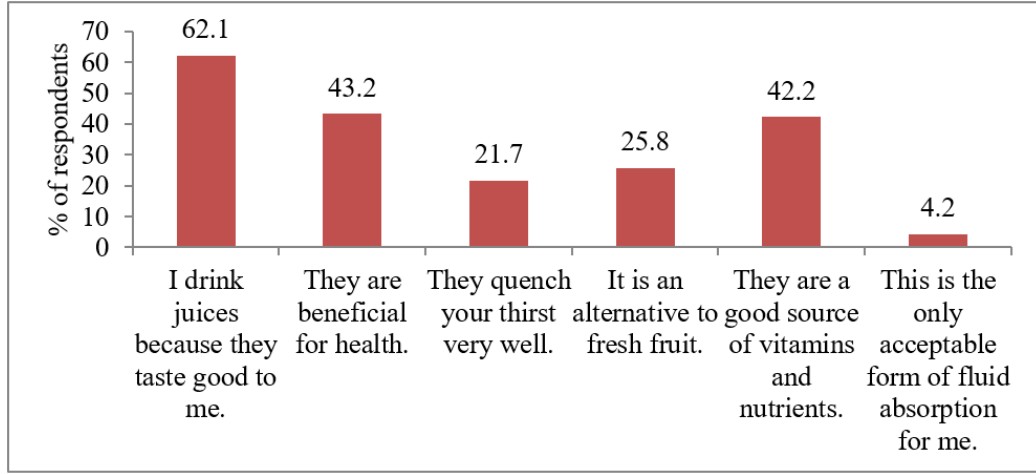

**Figure 3.** Reasons why respondents choose citrus juices.

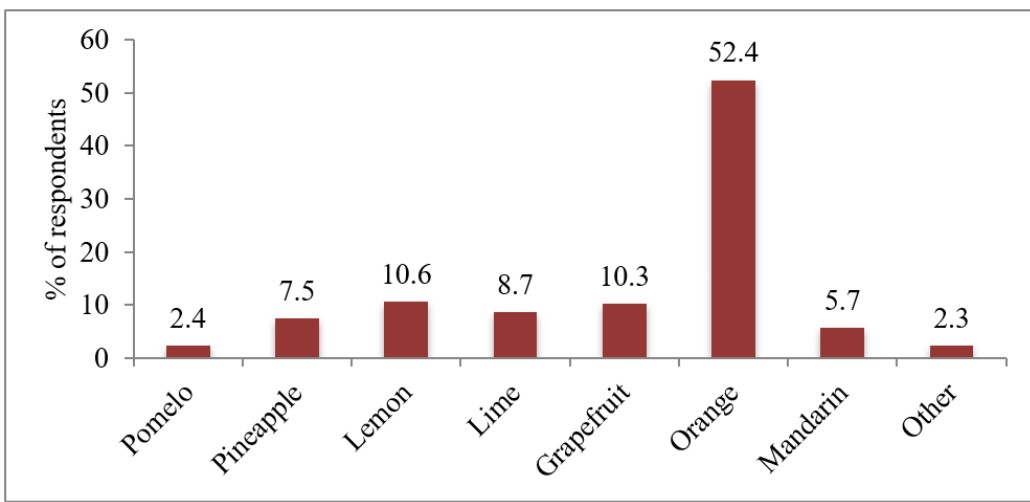

**Figure 4.** The most popular flavours of citrus fruit juices.

Women were more likely to choose orange juice than men. The relationship between the variables is statistically significant and characterised by weak strength of association (V Kramer = 0.214, $chi^2$ = 39.455, df = 7, $p$ = 0.000, p Monte Carlo = 0.000). Respondents in each age category were most likely to choose orange juice, although this was more pronounced among younger respondents (under 18, 18–24). The relationship between the variables is statistically significant and has a fairly weak strength of association (V Kramer = 0.133, $chi^2$ = 91.794, df = 42, $p$ = 0.000, p Monte Carlo = 0.000). Place of residence did not statistically significantly differentiate the most frequently chosen flavours of citrus juices (V Kramer = 0.094, $chi^2$ =38.327, df = 35, $p$ = 0.321, p Monte Carlo = 0.310). The taste of the most frequently chosen citrus fruit juices was not statistically significantly dependent on education level (V Kramer = 0.098, $chi^2$ = 25.039, df = 21, $p$ = 0.245, p Monte Carlo = 0.239).

Respondents were asked about their preference for the type of juice due to the form of production. Respondents were most interested in self-squeezed juices (40.5%). Less

than 30% of respondents preferred fruit juices reconstituted from concentrated juice. Direct juices made from raw juices, so-called NFC (not from concentrate), were chosen by only 13.5%. This kind of approach may suggest consumers' fear of faster spoilage of these beverages. According to Michalczyk et al. [30], these types of juices are not thermally preserved, so the processes leading to juice spoilage will occur faster in such products. Of the respondents, 18.6% indicated that the form of juice production does not matter to them (see Figure 5).

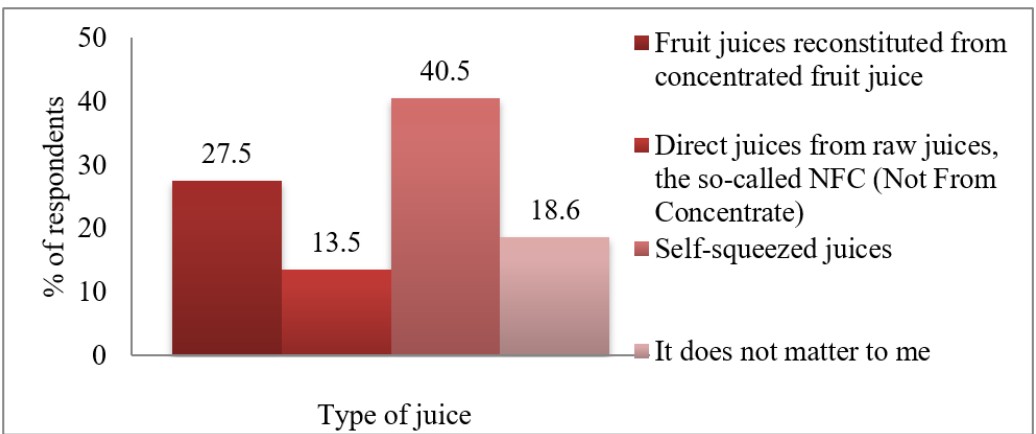

**Figure 5.** Preferences for choosing the type of citrus juices due to the form of production.

Women were more likely than men to choose self-squeezed juices, while men were more likely than women to drink direct juices from raw juice, the so-called NFC. The correlation coefficient is statistically significant and characterised by insignificant strength of association (V Kramer = 0.100, $chi^2$ = 8.643, df = 3, $p$ = 0.034, p Monte Carlo = 0.035). Respondents over 54 ages were more likely than younger respondents in this study to choose self-squeezed juices and NFC. Younger respondents were more likely than older respondents to drink fruit juices reconstituted from concentrated fruit juice. The correlation coefficient is statistically significant (V Kramer = 0.128, $chi^2$ = 42.691, df = 18, $p$ = 0.001, p Monte Carlo = 0.001). There was no statistically significant relationship with the size of the locality of residence (V Kramer = 0.059, $chi^2$ = 9.109, df =15, $p$ = 0.872, p Monte Carlo = 0.874) and education level (V Kramer = 0.078, $chi^2$ = 15.751, df = 9, $p$ = 0.072, p Monte Carlo = 0.070).

In another question addressed in the paper, respondents answered that they most preferred naturally cloudy juices—51.2%. A significant minority of respondents indicated that they were interested in concentrated fruit juices—14.8% and clarified juices—11.9% of respondents. Nearly a quarter of respondents answered that the consistency of the juice does not matter to them (see Figure 6).

Women were more likely than men to drink naturally cloudy juices while men were more likely than women to drink concentrated fruit juices. The relationship between the variables is statistically significant and has a weak strength of association (V Kramer = 0.100, $chi^2$ = 8.643, df = 3, $p$ = 0.034, p Monte Carlo = 0.035). Respondents over 25 were more likely to choose naturally cloudy juices than younger respondents, (those under 18 and those aged 18—24). However, it can be seen from the survey that both those under 18 and those over 65 did not attach importance to juice consistency. The relationship between the variables is statistically significant and shows insignificant strength of association (V Kramer = 0.108, $chi^2$ = 30.393, df = 18, $p$ = 0.034, p Monte Carlo = 0.034). The percentage of respondents answering the question "Considering appearance and texture, which citrus juices do you prefer the most?" was not statistically significantly dependent on place of residence (V Kramer = 0.092, $chi^2$ = 22.014, df = 15, $p$ = 0.107, p Monte Carlo = 0.103). Respondents characterised by higher education were more likely to reach for naturally cloudy juices. In contrast, for those characterised by lower education, this choice was not significant.

The coefficient of correlation is statistically significant and shows insignificant strength of association (V Kramer = 0.095, $chi^2$ = 23.175, df = 9, *p* = 0.006, p Monte Carlo = 0.006).

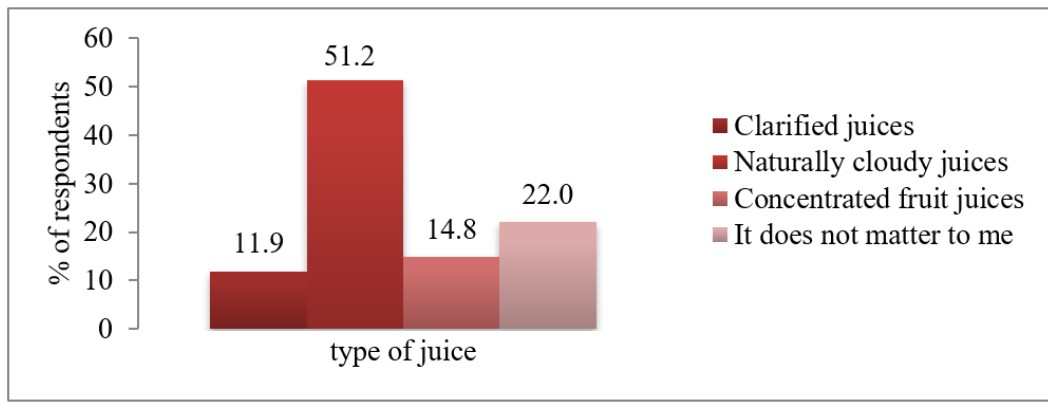

**Figure 6.** Respondents' preferences for choosing citrus juices due to texture.

A large proportion of respondents did not pay attention to the form of juice preparation. For 44% of respondents, it does not matter whether the juice is pasteurised or unpasteurised. Of the respondents, 36.8% chose unpasteurised juices, and 19.3% chose pasteurised juices (see Figure 7).

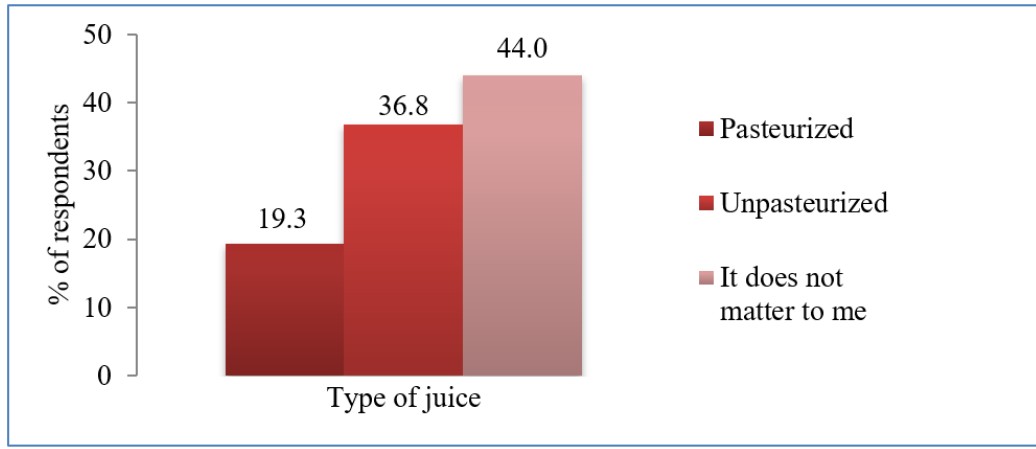

**Figure 7.** Selection of citrus juices due to the form of preparation (fixation) of the juice.

Unpasteurised juices were more often chosen by men, while for women, to a greater extent, the choice of the type of juice did not matter. On the result of the responses obtained regarding this issue, the coefficient of correlation is statistically significant and characterised by a weak strength of the relationship (V Kramer = 0.137, $chi^2$ = 16.075, df = 2, *p* = 0.000, p Monte Carlo = 0.000). Older respondents were more likely to choose unpasteurised juices, while for younger respondents to a greater extent it was irrelevant. The correlation coefficient is statistically significant and characterised by insignificant strength of association (V Kramer = 0.193, $chi^2$ = 64.329, df = 12, *p* = 0.000, p Monte Carlo = 0.000). The choice of the type of citrus fruit juice was not statistically significant taking into account the place of residence (V Kramer = 0.096, $chi^2$ = 15.856, df = 10, *p* = 0.104, p Monte Carlo = 0.101). Respondents with at least basic vocational education were more likely to choose unpasteurised juices, while for those with lower education to a greater extent the choice of citrus juice was not significant. The relationship between the variables is statistically significant and shows insignificant strength of association (V Kramer = 0.116, $chi^2$ = 23.070, df = 6, *p* = 0.001, p Monte Carlo = 0.001).

In the work presented, respondents were also asked to what extent the factors influencing their decisions to purchase citrus fruit juices. The most important issue that was mentioned by respondents was the type of fruit from which the juice was made. The sum of responses indicated as "very important and important" was 80%, for 13.2% it was

"unimportant", and "not important and definitely not important" for 6.7% of respondents. Sensory sensations "are very important and important" for 66.7% of respondents, for 24% "it is not important", and "not important and definitely not important" for 9.3%. An important aspect with which respondents associated the consumption of juices was the nutritional value and, more specifically, the vitamin content of these drinks. "Very important and important" for 77.9% of respondents, it was "unimportant" for 14.7%, and "not important and definitely not important" for 7.4%. A much lower result was obtained by Sicińska and Pelc in their study [31], where only 46% of respondents indicated the addition of vitamins or minerals as a factor influencing the purchase of juices. The third factor influencing the purchase of juice was economic—the price of the product, which was "very important and important" to 71.4% of respondents, 19.3% did not pay attention to it, and 9.4% of respondents did not care about the price. Factors such as sugar content, texture and type of intermediate product were "very important and important" to just over 2/3 of respondents, 69.7%, 66.9%, and 64.9% of respondents, respectively. The least important factors in the selection of juices were the attractiveness of the packaging, the country of origin of the product, and the brand, for which the sum of the indications of the following responses of respondents "unimportant", "not important", and "definitely not important" were, respectively: 66.2%, 63%, and 56.6%.

EFSA's 2019 "Eurobarometer" survey shows that the most important factor for Europeans is where the product comes from (53%). The next factor in choosing juice is price (51%), followed by food safety (50%) and taste (49%). Polish people, on the other hand, identified taste as the most important factor (58%), followed by price (53%), and nutrient content (48%) [32].

The presented study also noted that among other factors indicated by respondents that guide their choice of citrus fruit juices were organic certification, eco-friendly packaging, calorie content of the juice, size and type of packaging, advertising, and availability in stores.

The most frequently cited citrus fruit juice producer was the company "O" (22.2%). The result may prove that respondents associate juices with a particular brand. Similar observations were confirmed by the authors Goldsmith et al. [33], who indicated that brand choice matters to respondents. In second place, respondents indicated the juice produced by company "E" (18.9%), followed by the juice producer "C" (11%), and "F" (7.5%). For the remaining companies, respondents' interest remained at no more than 5.5% (see Figure 8).

Gender did not statistically significantly differentiate the citrus producer of the most frequently chosen juices (V Kramer = 0.164, $chi^2$ = 23.066, df = 18, $p$ = 0.188, p Monte Carlo = 0.182). Age of respondents statistically significantly differentiated choice of citrus juice producer. The strength of the relationship between the variables was V Kramer = 0.220. It can be observed that "E" company was the producer most often chosen by the oldest respondents (65 years and older), while "O" company was chosen by younger respondents (under 18 and aged 18–24) (V Kramer = 0.220, $chi^2$ = 249.324, df = 108, $p$ = 0.000, p Monte Carlo = 0.000). The choice of citrus juice company was not statistically significantly dependent on the place of residence (V Kramer = 0.154, $chi^2$ = 101.681, df = 90, $p$ = 0.188, p Monte Carlo = 0.176). "O" company was most often chosen among respondents with primary education, while respondents with higher levels of education chose "O" company and "E" company to a similar extent. The relationship between variables was statistically significant and characterised by insignificant strength of association (V Kramer = 0.187, $chi^2$ = 90.230, df = 54, $p$ = 0.001, p Monte Carlo = 0.001).

### 3.2. Organoleptic Evaluation of Commercial Juice and Juices Made with a Juicer or Squeezer

Fruit flavour and nutritional characteristics are key quality traits and one of the main factors influencing consumer preference [34]. The organoleptic evaluation included self-made juices and commercial juice "O", which was indicated in the survey as the most attractive and appreciated by the respondents. Such parameters were evaluated: the colour, aroma, taste, and appearance of the juices. The highest average rating for the

aforementioned parameters was given to EW juice with 4.3, followed by ES juice with 4.1 (see Table 1). The evaluation of commercial juice was a rating similar to that of ES juice (4.0). The weakest rating was given to HS juice with an average rating of 2.8. Commercial orange juice "O" received a similar rating (4.03) in an assessment of orange juice quality conducted by Pyryt and Wilkowska [35]. This study evaluated orange juices from three Polish producers: Hortex, Fortuna, and Tymbark.

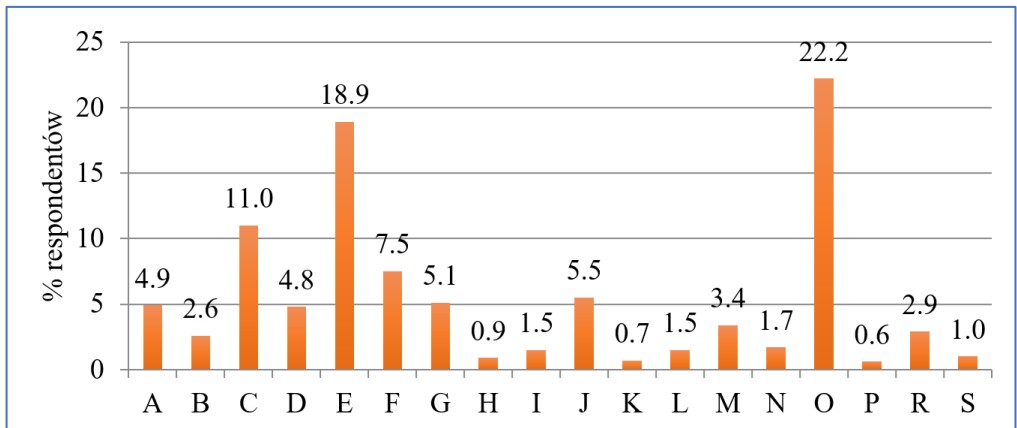

**Figure 8.** Percentage of respondents who chose a company's citrus juice. Legend: A–S the type of the juice produced by different Polish companies.

**Table 1.** Organoleptic evaluation of produced juices and commercial juice.

| Type of Juice | Colour | Fragrance | Taste | Appearance | Average |
|---|---|---|---|---|---|
| ES | 4.0 | 4.5 | 4.1 | 3.6 | 4.1 |
| EW | 4.6 | 4.1 | 4.5 | 3.8 | 4.3 |
| HS | 3.9 | 2.5 | 1.6 | 3.4 | 2.8 |
| HW | 4.8 | 4.0 | 3.1 | 3.6 | 3.9 |
| O | 3.4 | 3.9 | 4.0 | 4.8 | 4.0 |

Legend: Squeezed using a juicer on oranges originating in Spain (HS). Squeezed using a juicer on oranges originating in Egypt (ES). Squeezed using a squeezer on oranges originating in Spain (HW). Squeezed using a squeezer on oranges originating in Egypt (EW). Commercial juice coming from Polish company (O).

The juices squeezed with a squeezer had the most intense colour. Their colour was intensely orange. The colour of juice from oranges from Egypt received a rating of 4.6, and oranges from Spain received 4.8. The colour of juices squeezed with a juicer was slightly yellow, the rating for oranges from Egypt was 4.0, and for oranges from Spain 3.9. Commercial juice "O" had the brightest yellow colour, the average rating for the colour of this juice was 3.4. According to Li et al. [21], aroma index is one of the important qualities of citrus juice, and this indicates that the quality may be judged from the changes of these compounds. Aroma is generated by a complex mixture of volatiles emitted by the fruit; however, even if hundreds of volatiles are detected in most fruits, a small subset of them is thought to be responsible for their distinctive fragrance [36–38]. Aroma was rated highest for ES juice—4.5. There was little difference in the ratings for the aroma of EW and HW juice and commercial "O" juice, for which the ratings were 4.1, 4.0, and 3.9, respectively. The worst rating for juice aroma was given to HS juice—2.5. This juice had an unpleasant, slightly bitter aroma. Monoterpenes and sesquiterpenes greatly contribute to floral emissions and the aroma of several fruits, including citrus [39]. The taste of HS juice was rated the lowest of all—1.6. This juice was bitter, slightly astringent, and generally tasteless. HW juice was rated 3.1. It was characterised by a bitter and tarty taste, although much less than HS juice. The taste of Egyptian orange juices was rated highly, for EW juice—4.5, and for ES juice—4.1. These juices were sweet and tasty. Commercial juice "O" was characterised as sweet and sour, and as a result it received a rating of 4.0. Similar to a study by Jönsson and Nybom [40], taste and flavour, associated with optimum juiciness, are

considered the most important quality traits. The appearance of the juice was rated highest for commercial "O" juice—4.8. In this juice no pulp particles were perceptible at all, the texture was homogenous and uniform. In the other juices, pulp particles were perceptible, although there were slightly fewer in juices squeezed with a juicer.

Consumer preferences are currently forcing growers to pay attention to fruit quality traits such as flavour and nutritional value [41,42]. For this reason, fruit metabolism has become an obvious target for the production of better-tasting and healthier fruits [43]. While primary metabolites, such as sugars and acids, directly influence fruit taste, secondary metabolites, such as polyphenols, terpenoids, and volatiles, are also responsible for their quality by being involved in their aroma, colour, and health-promoting characteristics [34].

### 3.3. Physicochemical Evaluation of the Commercial Juice and Juices Made with a Juicer or Squeezer

Fruit quality represents an essential breeding objective, as consumers' expectations are constantly growing [44,45] and this influences the quality of the resulting juices. Quality is a complex trait, dependent on morphologic and organoleptic characteristics of fruit (size, shape, skin colour, flavour, taste, juiciness, crispness, firmness, etc.), but at the same time influenced by agrotechnical practice, biochemical processes, and nutritional richness of fruits [46–48]. Physicochemical analysis of the resulting juices was also carried out (see Table 2. According to the *Good Manufacturing Practice Guide for the Juicing Industry* [49], the relative density of orange juices from fresh raw material should be 1.040 $g/cm^3$, and for juice reconstituted from concentrated juice it should be 1.045 $g/cm^3$. For fresh juices, the density obtained was close to the value of 1.040 $g/cm^3$. The highest densities were obtained for juices squeezed in a squeezer, both for juice from oranges from Egypt and Spain, with values of 1.042 $g/cm^3$ and 1.043 $g/cm^3$, respectively, which may have been due to the greater presence of pulp particles in the juice. Slightly lower values were noted for juices squeezed in a juicer, for which the values were 1.038 $g/cm^3$ for orange juice from Egypt, and 1.034 $g/cm^3$ for orange juice from Spain. In the case of the commercial juice, the density was lower and amounted to 1.041 $g/cm^3$ (see Table 2).

**Table 2.** Physicochemical evaluation of produced juices and commercial juice.

| Parameters | ES | EW | HS | HW | O |
|---|---|---|---|---|---|
| Density | 1.038 | 1.042 | 1.034 | 1.043 | 1.041 |
| pH | 4.2 | 4.1 | 4.2 | 4.4 | 3.9 |
| Sugar content | 4.2 | 4.1 | 4.2 | 4.4 | 3.9 |
| Dry matter content (%) | 10.88 | 10.32 | 8.75 | 7.25 | 10.75 |

Legend: Squeezed using a juicer on oranges originating in Spain (HS). Squeezed using a juicer on oranges originating in Egypt (ES). Squeezed using a squeezer on oranges originating in Spain (HW). Squeezed using a squeezer on oranges originating in Egypt (EW). Commercial juice coming from Polish company (O).

Similar values were obtained for the pH of the juices tested. For fresh juices, the pH ranged from 4.1 to 4.4, while for commercial juice it was the lowest at 3.9 (see Table 3). In a study published by Michalak-Majewska et al. [50], a pH of 3.69 ± 0.39 was obtained for commercial orange juice, which was lower than the pH of all juices tested. Parish [51] reported that the pH of freshly squeezed orange juice varied between 3.3 and 4.3, which agrees with our results.

**Table 3.** Vitamin C content in orange juices before and after pasteurisation.

| Type of Juice | Content of Vitamin C * (mg/100 cm$^3$) | Content of Vitamin C ** (mg/100 cm$^3$) | Content of Vitamin C * (%) |
|---|---|---|---|
| ES | 33.7 | 32.9 | 2.4 |
| EW | 30.6 | 30.5 | 0.1 |
| HS | 49.1 | 48.7 | 0.8 |
| HW | 34.6 | 33.7 | 2.6 |
| O | 29.8 | 26.6 | 10.7 |

Legend: Squeezed using a juicer on oranges originating in Spain (HS). Squeezed using a juicer on oranges originating in Egypt (ES). Squeezed using a squeezer on oranges originating in Spain (HW). Squeezed using a squeezer on oranges originating in Egypt (EW). Commercial juice from Polish company (O). * before pasteurisation; ** after pasteurisation at 72 °C, time 10 s.

The highest sugar level for commercial orange juice "O" and ES juice was 10.2 g. According to the commercial juice manufacturer "O", the content of all sugars is 10 g. A slightly higher sugar level was determined for EW juice—10.15 g. The lowest values were recorded for HS juice—8 and HW juice—7 (see Table 2), which is consistent with the results of the organoleptic evaluation, in which juices from oranges from Spain were rated as the least sweet. The determination of sugars in juices was also handled by Lebiedzinska et al. [52], who used high-performance liquid chromatography (HPLC). The author indicated that in orange juice the content of fructose was $1.82 \pm 0.20$ g/100 cm$^3$, glucose $1.56 \pm 0.20$ g/100 cm$^3$, sucrose $3.14 \pm 0.13$ g/100 cm$^3$, providing a total sugar content of $6.51 \pm 0.38$ g/100 mL. From the results presented in this work for commercial orange juice, the content of fructose was $2.22 \pm 0.08$ g/100 cm$^3$, glucose $2.04 \pm 0.16$ g/100 cm$^3$, and sucrose $3.69 \pm 0.09$ g/100 cm$^3$; the total content of sugars was $7.97 \pm 0.24$ g/100 cm$^3$. Comparing these data with the information provided by Lebiedzinska et al. [52], the amount of determined sugars in the presented work was higher.

An important parameter in the characterisation of juices is the dry-matter content. Comparable values of dry-matter content were obtained for ES juices—10.88%, commercial orange juice 10.75%, and squeezed orange juice from Egypt 10.32%. Significantly lower values of dry matter were noted for orange juice from Spain, squeezed in a juicer—8.75%, and a squeezer—7.25% (see Table 2). The above data differentiate the water content of the juices depending on where the fruit came from. Smaller differences were observed in juices obtained by the squeezing method in the squeezer and juicer.

The vitamin C content of juices, according to the *Good Production Practice Guide for the Juicing Industry* [49], should be a minimum of 200 mg/L. From the research conducted in the work presented, more vitamin C was contained in juices from oranges originating in Spain than in juices from oranges originating in Egypt. Juices from oranges squeezed with a juicer showed more vitamin C than juices squeezed with a squeezer (see Table 3). The vitamin C content of commercial juice was significantly higher than that stated by the manufacturer on the package (23.8 units). This inconsistency may be due to the deliberate administration of higher amounts of vitamin C during juice production. As reported by Polydera et al. [53], the content of vitamin C during the production process may decrease due to its perishability, hence higher amounts are given so that the declared amount is not underestimated. In the work presented it was observed that the vitamin C content decreased slightly after pasteurisation at 72 °C. Baldwin et al. [54] reported that the heating process in thermal pasteurisation might have caused an increase in the total acids. The main reason of the increase of the total acid in juice might be the generation of acids from carbohydrate according to the oxidation [55]. This did not agree with our results. The biggest difference between the vitamin C content in the juice before and after pasteurisation was recorded for commercial juice (3.2 mg/100 cm$^3$). For orange juice squeezed with a squeezer, for both Egyptian and Spanish juice, the vitamin C content was virtually unchanged (see Table 4). Studies of the effect of temperature on the physicochemical changes of orange juice were also conducted by Yuk et al. [56]. Storage time, temperature and microbial contamination significantly affected the flavour of fruit juice [57]. Pasteurisation was carried out at 70 °C for 7.2 s, and there was no significant reduction in vitamin C either. Petruzzi et al. [58] obtained larger changes in vitamin C content, who conducted fixation at a higher temperature (90 °C/1 min). This confirms that higher temperature causes decomposition and greater loss in the content of this vitamin.

The colour of food products and juices is one of the most important determinants of quality. The colour of the juice was measured according to the CIE L*a*b* system, which represents colour in three-dimensional space. The colour of the juices was measured after squeezing and after pasteurisation. The L*a*b* values and the colour difference between fresh and pasteurised juice are shown in Table 5. L* stands for brightness; according to this value each colour can be assimilated to a greyscale between 0 (black) and 100 (white). Positive values of the coefficient a* correspond to red, and negative values correspond to green. For the coefficient b*, positive values correspond to blue colours, and negative

values correspond to yellow colours. ΔE is the colour difference between juice before and after pasteurisation [59].

**Table 4.** L*a*b* colour rating and colour difference ΔE before (1) and after pasteurisation (2).

| Type of Juice | | C* | L* | a* | b* | ΔE |
|---|---|---|---|---|---|---|
| ES | 1 | 18.97 | 36.90 | −3.62 | 18.62 | 3.8 |
| | 2 | 21.26 | 39.93 | −4.08 | 20.86 | |
| EW | 1 | 16.83 | 33.37 | −2.09 | 16.70 | 10.3 |
| | 2 | 19.03 | 43.40 | −3.03 | 18.79 | |
| HS | 1 | 31.20 | 42.87 | −0.95 | 31.19 | 3.8 |
| | 2 | 34.88 | 43.84 | −1.53 | 34.85 | |
| HW | 1 | 21.17 | 36.63 | −0.56 | 21.16 | 5.6 |
| | | 26.15 | 39.02 | −1.45 | 26.11 | |
| O | 1 | 19.49 | 40.23 | −3.92 | 19.09 | 4.2 |
| | 2 | 21.23 | 44.02 | −4.07 | 20.84 | |

Legend: Squeezed using a juicer on oranges originating in Spain (HS). Squeezed using a juicer on oranges originating in Egypt (ES). Squeezed using a squeezer on oranges originating in Spain (HW). Squeezed using a squeezer on oranges originating in Egypt (EW). Commercial juice from Polish company (O).

**Table 5.** Weight of oranges, the weight of juice, and percent efficiency.

| Type of Juice | Weight of Oranges (g) | Weight of the Obtained Juice (g) | Efficiency (%) |
|---|---|---|---|
| ES | 1001.36 | 499.90 | 49.9 |
| EW | 1177.11 | 540.00 | 45.9 |
| HS | 962.50 | 475.64 | 49.4 |
| HW | 951.70 | 400.55 | 42.1 |

Legend: Squeezed using a juicer on oranges originating in Spain (HS). Squeezed using a juicer on oranges originating in Egypt (ES). Squeezed using a squeezer on oranges originating in Spain (HW). Squeezed using a squeezer on oranges originating in Egypt (EW).

After pasteurisation, a colour change toward higher L* values was observed for each juice (samples became lighter). The values of the coefficient a* decreased, suggesting changes toward green colour. In turn, an increase in the value of the coefficient b* confirmed a change in colour toward yellow (see Table 4). Similar results were obtained in the study of Gabriel and Azanza [60], where the sample's brightness increased after heating. In general, it was found that the colour of the juices was much brighter than before pasteurisation. The difference in colour after pasteurisation compared to that of fresh juice was distinct (ΔE > 3). In addition to sensory perception, colour informs the product's nutritional value. Colour changes can be considered as the degree of carotenoids lost during thermal fixation [61]. The colour difference can also be indicative of changes taking place in food products [62]. The quali–quantitative evaluation and the improvement of the levels of plant bioactive secondary metabolites are increasingly gaining consideration by growers, breeders, and processors, particularly in those fruits and vegetables that, due to their supposed health-promoting properties, are considered "functional" [63].

The most juice was obtained from oranges, which were squeezed using a juicer (nearly 50% by weight of oranges). In contrast, for juices squeezed in a juicer, this yield was less than 50%.

Considering the type of juice production with a squeezer and the method with a juicer, the results clearly confirm that electric juicing devices are more efficient and allow for the production of more juice than squeezing by hand (see Table 5).

### 3.4. Cost Analysis of Orange Juice

This study also conducted a brief overview of the cost of producing orange juice and referred to the price of commercial juice "O". It turned out that the price of store-bought orange juice is significantly lower than that of self-made juice. Commercial juice is more cost-effective; it costs three times less than self-squeezed juice (see Table 6).

**Table 6.** Cost of oranges and orange juices.

| Type of Product | Cost |
|---|---|
| Oranges | 1.27 €/kg |
| Squeezed orange juice | 2.54 €/L |
| Tymbark orange juice | 0.90 €/L |

## 4. Summary and Conclusions

In the survey, respondents indicated that they were most likely to choose orange juice (52.4%).

Juices were consumed preferably one to three times a week. Respondents were mainly guided by taste. The following factors were also crucial to respondents: health benefits, the presence of vitamins and nutrients, and price.

In general, respondents preferred self-squeezed juices and naturally cloudy juices and were not interested in whether the juice had been fixed or not.

Organoleptic analysis showed that orange juice made from fruit originating in Egypt scored higher than orange juice derived from fruit originating in Spain. In addition, the former-mentioned juice gained the highest vote in terms of taste, while the latter juice was characterized by a bitter taste. The commercial juice "O", which was indicated in the survey by respondents as the best, differed significantly from self-squeezed juices, primarily in colour.

The choice of juice extraction device was found to affect the juice's organoleptic characteristics. Juice from oranges squeezed in a squeezer had a more intense, distinct colour than juice obtained with a juicer. Juice obtained with a juicer contained less pulp than juice squeezed with a squeezer. A higher efficiency was achieved when the juice was obtained using the juicer.

Commercial juice "O" met all the requirements, only the density was slightly lower than that required for juices. In the case of orange juice from Egypt, titratable acidity was a parameter that differed from the requirements for juices. The same parameter and the content of soluble substances were not met for the juice obtained from oranges originating in Spain. Determined vitamin C in all tested juices was determined at the required level for citrus fruit juices.

The commodity assessment of juices assessed In the study remains high, so it can be concluded that the produced drinks are a good, nutritious, and safe product for consumers.

The work also confirmed that consumers regard juices as an essential part of their diet. Introducing into the diet as well as not excluding such products during the COVID-19 period, in the opinion of the authors, is exceptionally reasonable in the context of vitamin C present in juices. Related to the above statement, growing consumer awareness of preventive medicine and immune enhancement places higher demands on these products, so future research into the bioavailability of functional compounds and their interactions is warranted. Perhaps these studies will more clearly define the boundaries between long-term commercial juices and short-term juices made with a squeezer or juicer.

**Author Contributions:** Conceptualization, M.K. and J.K.; methodology, M.K and J.K.; software, B.K.; validation, M.F. and A.Z.; formal analysis, M.F. and A.Z.; investigation, J.K.; resources, J.K.; data curation, J.K and M.K.; writing—original draft preparation, J.K and M.K.; writing—review and editing, M.F. and B.K.; supervision, A.Z. All authors have read and agreed to the published version of the manuscript.

**Funding:** This research received no external funding.

**Institutional Review Board Statement:** Not applicable.

**Informed Consent Statement:** Not applicable.

**Data Availability Statement:** Not applicable.

**Acknowledgments:** This research was supported by the Kazimierz Pulaski University of Technology and Humanities in Radom, Poland.

**Conflicts of Interest:** The authors declare no conflict of interest.

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
