# Peer review of "Quality Assessment of Natural Juices and Consumer Preferences in the Range of Citrus Fruit Juices"

_applsci, doi:10.3390/app13020765_

Round 1
Reviewer 1 Report
In general, the text should be revised in several parts as reported in the detailed comments.

Reviewer 2 Report
Dear Authors,
your study is not suitable for the Issue you applied for. Moreover, I don't think it brings any valuable information to the scientific community. You haven't followed how taste has changed over the years. You just did a snapshot of the taste of a small number of people from homogenized nationality.
Author Response
Radom, Poland, October 24, 2022
manuscript number : applsci-1977747
Title: QUALITY ASSESSMENT OF NATURAL JUICES AND CONSUMER PREFERENCES IN THE RANGE OF CITRUS FRUIT JUICES
Authors: Małgorzata KOWALSKA, Justyna KONOPSKA, Melania FESZTEROVA, Anna ZBIKOWSKA, Barbara KOWALSKA;
Dear Editor,
Dear Reviewer,
Thank you very much for the constructive comments and suggestion. According to Reviewer suggestions all corrections and information were placed in the manuscript. The modifications of the manuscript were highlighted (in red colour). Answers and references to all comments and suggestions are marked with a slanted font.
Reviewer 2
Dear Authors,
your study is not suitable for the Issue you applied for. Moreover, I don't think it brings any valuable information to the scientific community. You haven't followed how taste has changed over the years. You just did a snapshot of the taste of a small number of people from homogenized nationality.
Answer: Thank you for the suggestion to do a study to evaluate the change in juice taste over the years. I think it would indeed be interesting especially in terms of determining the many determinants that may have influenced it. Referring to our research, on the other hand, I must reply that it dealt with a completely different topic. Mainly the research focused on in the direction of determining behavior, consumer preferences when choosing citrus fruit juices. We also included in our research a short research episode for the most attractive juice chosen by consumers, as well as juices that can be called „instant” i.e. manufactured and directly ready to drink. In our opinion, consumer preferences are a key issue for citrus juice producers. These indications, choices determine new flavors or new products. Determining consumer preferences often becomes a reason for manufacturers to produce new or modified products. Food or juice producers often use such data to make a new juice produced to meet consumer needs, which of course is strongly related to the sales policy of such a product.

Reviewer 3 Report
Comments
The manuscript " Quality Assessment of Natural Juices and Consumer Prefer-2 ences in the Range of Citrus Fruit Juices " represents good work. However, the manuscript needs more analyses to support the author’s findings and be considered the manuscript for publication.
1. The authors need to use better statistical and correlation analysis to show the data properly and more high-quality shape.
2. Some data legends are missing in Figures 5 and 7.
3. Discussion section needs more explanation.
4. The language needs more revision.

Author Response
Radom, Poland, October 24, 2022
manuscript number : applsci-1977747
Title: QUALITY ASSESSMENT OF NATURAL JUICES AND CONSUMER PREFERENCES IN THE RANGE OF CITRUS FRUIT JUICES
Authors: Małgorzata KOWALSKA, Justyna KONOPSKA, Melania FESZTEROVA, Anna ZBIKOWSKA, Barbara KOWALSKA;
Dear Editor,
Dear Reviewer,
Thank you very much for the constructive comments and suggestion. According to Reviewer suggestions all corrections and information were placed in the manuscript. The modifications of the manuscript were highlighted (in red colour). Answers and references to all comments and suggestions are marked with a slanted font.
Reviewer 3
The manuscript " Quality Assessment of Natural Juices and Consumer Prefer-2 ences in the Range of Citrus Fruit Juices " represents good work. However, the manuscript needs more analyses to support the author’s findings and be considered the manuscript for publication.
- The authors need to use better statistical and correlation analysis to show the data properly and more high-quality shape.
Answer: Thank you for this comment, I think that at this stage the inclusion of another analysis would spoil the shape of the work that was created after the introduction of the analysis used in the paper. Although I do not exclude that perhaps this is a way to present the data in a different, better perhaps more clear way.
- Some data legends are missing in Figures 5 and 7.
Answer: I has been corrected
- Discussion section needs more explanation.
Answer: Thank you for this comment. The more explanation has been introduced
- The language needs more revision.
Answer: The revision has been done

Reviewer 4 Report
Comments
These are my remarks and suggestions:
Page 2. Instead of 2.1. Survey questionnaire I suggest 2.1. Questionnaire design.
Page 3. What were the selection criteria for juices from table 1 for further physico-chemical and sensory analysis?
Lines 89-91 are repeated. See lines 82-85
Lines 82-93 and subsection 2.2. should be renamed 2.2. Sensory evaluation
Lines 251-252. Please rephrase
Line 290 “juice fixation” please use another term for fixation
Figure 8. Are only the producers different? The juice compositions are the same for all the producers? Please supply more information on juices A-S
Line 367“produced juices and commercial juice.” What is the difference between produced and commercial? Why did the authors compare only the commercial?
In the Conclusions section, add more information on the practical application of the results obtained during the research and specify this study's novelty.
The comparison between freshly obtained juices (made by the authors) and commercial ones is not entirely justified. The juices are difficult to compare, only based on one criterion shown by the survey as the most attractive and appreciated by the respondents. Overall, it is not easy to follow the logical flow of the manuscript.
Author Response
Radom, Poland, October 24, 2022
manuscript number : applsci-1977747
Title: QUALITY ASSESSMENT OF NATURAL JUICES AND CONSUMER PREFERENCES IN THE RANGE OF CITRUS FRUIT JUICES
Authors: Małgorzata KOWALSKA, Justyna KONOPSKA, Melania FESZTEROVA, Anna ZBIKOWSKA, Barbara KOWALSKA;
Dear Editor,
Dear Reviewer,
Thank you very much for the constructive comments and suggestion. According to Reviewer suggestions all corrections and information were placed in the manuscript. The modifications of the manuscript were highlighted (in red colour). Answers and references to all comments and suggestions are marked with a slanted font.
Reviewer 4
These are my remarks and suggestions:
- Page 2.Instead of 1. Survey questionnaire I suggest 2.1. Questionnaire design.
Answer:The change has been made.
- Page 3.What were the selection criteria for juices from table 1 for further physico-chemical and sensory analysis?
Answer: The clarifying sentence has been added
- Lines 89-91are repeated. See lines 82-85
Answer: The correction has been done.
- Lines 82-93and subsection 2.2. should be renamed 2. Sensory evaluation
Answer:The correction has been done.
- Lines 251-252. Please rephrase
Answer: The sentence has been corrected.
- Line 290 “juice fixation” please use another term for fixation
Answer: It has changed.
- Figure 8. Are only the producers different? The juice compositions are the same for all the producers? Please supply more information on juices A-S
Answer: This question was not about the composition of the juices, but rather about the recognition of the producer, or rather, to indicate the citrus juice producer that is most recognized by the respondents. Which, in their opinion, produces the tastiest juices. This figure shows that one company is clearly chosen by the majority of respondents, which, of course, may suggest that respondents point to the high quality of juices from this producer.
- Line 367“produced juices and commercial juice.” What is the difference between produced and commercial? Why did the authors compare only the commercial?
Answer:The correction has been introduced into the manuscript. The commercial juice was this product which has been chosen by respondents. It was only one juice "O"
- In the Conclusions section, add more information on the practical application of the results obtained during the research and specify this study's novelty.
Answer: A short paragraphs have been added
- The comparison between freshly obtained juices(made by the authors) and commercial ones is not entirely justified. The juices are difficult to compare, only based on one criterion shown by the survey as the most attractive and appreciated by the respondents. Overall, it is not easy to follow the logical flow of the manuscript.
Answer:.The message of our work was to show consumer preferences. Also, we wanted to find out whether, with such a wide variety of juices on the Polish market, there is any juice that respondents clearly indicate as the best. It turned out that there is. So we tried to compare the juice indicated by respondents to juices that are directly produced and directly consumed, that is, juices that are not stored, not pasteurized, that is, not subjected to any processing. It turned out that in sensory evaluation the juice chosen by respondents in the survey was rated similarly to juice made directly with a juicer or squeezer. This confirms that consumers can trust a given producer to produce reproducible juice with unchanging determinants that characterize its quality.
The presented paper attempts to fill the gap in information about juices regarding nutrition, taste as well as culinary opinions. It is based on an integrated approach, considering not only nutrition but especially nutritional changes and food science applications in juice sector. In addition, it presents the results of a survey on consumer preferences and a new perspective on the sensory properties of juices. However, like other food sectors, the juice sector has begun to grow through several technology modifications, increased assortment and marketing efforts.

Round 2
Reviewer 1 Report
The authors revised the manuscript and improved some parts respecting the detailed comments
Author Response
Dear Reviewer,
Thank you for your comments and accepted our new version of the manuscript...
Reviewer 2 Report
Although the Authors put evident efforts to improve their manuscript it does not fit the purpose of the Issue. The Conclusion does not intelligibly reflect the main results. It contains references that are inappropriate and unnecessary (regarding the purpose of the section).
Author Response
Dear Reviewer,
Thank you for your comment. The conclusions have been corrected.
Reviewer 4 Report
I agree with the modifications and corrections made by the authors
Author Response
Dear Reviewer,
Thank you for your acceptation of modified version of the manuscript.